# Heritability of Cardiothoracic Ratio and Aortic Arch Calcification in Twins

**DOI:** 10.3390/medicina57050421

**Published:** 2021-04-27

**Authors:** Zsofia Jokkel, Bianka Forgo, Christopher Hani-Gaius Ghattas, Marton Piroska, Helga Szabó, David L. Tarnoki, Adam D. Tarnoki, Sooji Lee, Joohon Sung

**Affiliations:** 1Medical Imaging Centre, Semmelweis University, 78/A Üllői Street, 1082 Budapest, Hungary; christopher.gha@hotmail.com (C.H.-G.G.); piroskamarton94@gmail.com (M.P.); szabo.helga.se@gmail.com (H.S.); tarnoki4@gmail.com (D.L.T.); tarnoki2@gmail.com (A.D.T.); 2Department of Radiology, Faculty of Medicine and Health, Örebro University, Fakultetsgatan 1, 702 81 Örebro, Sweden; fbia021@gmail.com; 3Genome and Health Big Data Laboratory, Department of Public Health, Graduate School of Public Health, Seoul National University, 1 Gwanak-ro Gwanak-gu, Seoul 08826, Korea; slee53@snu.ac.kr (S.L.); jsung@snu.ac.kr (J.S.); 4Institute of Health and Environment, Seoul National University, 1 Gwanak-ro Gwanak-gu, Seoul 08826, Korea

**Keywords:** twin study, heritability, aortic calcification, cardiothoracic ratio, atherosclerosis

## Abstract

*Background and Objectives*: Aortic arch calcification (AoAC) is associated with a variety of cardiovascular complications. The measurement and grading of AoAC using posteroanterior (PA) chest X-rays are well established. The cardiothoracic ratio (CTR) can be simultaneously measured with PA chest X-rays and used as an index of cardiomegaly. The genetic and environmental contributions to the degree of the AoAC and CTR are not well understood. The purpose of this study was to investigate the effect of genetics and environmental factors on the AoAC and CTR. *Materials and Methods*: A total of 684 twins from the South Korean twin registry (261 monozygotic, MZ and 81 dizygotic, DZ pairs; mean age 38.6 ± 7.9 years, male/female = 264/420) underwent PA chest X-rays. Cardiovascular risk factors and anthropometric data were also collected. The AoAC and CTR were measured and graded using a standardized method. A structural equation method was used to calculate the proportion of variance explained by genetic and environmental factors behind AoAC and CTR. *Results*: The within-pair differences were low regarding the grade of AoAC, with only a few twin pairs showing large intra-pair differences. We found that the thoracic width showed high heritability (0.67, 95% CI: 0.59–0.73, *p* = 1). Moderate heritability was detected regarding cardiac width (0.54, 95% CI: 0.45–0.62, *p* = 0.572) and CTR (0.54, 95% CI: 0.44–0.62, *p* = 0.701). *Conclusions*: The heritable component was significant regarding thoracic width, cardiac width, and the CTR.

## 1. Introduction

Cardiovascular disease due to atherosclerosis is the leading cause of mortality worldwide [1]. Atherosclerosis is a chronic vascular condition starting with fatty streak line that progresses over time into fibrous plaques. These plaques may eventually rupture, leading to thrombosis or stenosis. Numerous pathological processes have been identified as key factors in atherosclerosis development, including lipid retention, oxidation, and regulation of blood lipid levels [2]. The clinical manifestation of atherosclerosis could lead to different cardiovascular conditions such as ischemic heart disease, myocardial infarction, or aneurysms, while it can also be the cause of life-threatening complication including dissection or penetrating ulceration [3,4].

The cardiothoracic ratio (CTR) has been identified as an effective and easily accessible method for heart size estimation and cardiomegaly screening [5]. It is obtained by measuring the largest distance between the right and left edges of the heart and thorax on posteroanterior (PA) chest X-rays and dividing the cardiac size by the thoracic size. A value of 50% is generally accepted as the upper limit of normal CTR [6].

Several studies examined the possibility of aortic calcification and CTR as predictors of cardiovascular disease. Irribaren et al. found an association between aortic arch calcification and an increased risk for coronary heart disease of 27% in men and 22% in women, while they discovered an independent association with a 46% increased risk of ischemic stroke in women [7]. The CTR may also be used for the assessment of cardiomegaly [5].

Advanced imaging methods are critical in the diagnostics of cardiomegaly and aortic atherosclerosis due to the severe complications and their elevated morbidity and mortality rates of these conditions. Chest X-rays and transthoracic echocardiography are the most common imaging modalities used in cardiomegaly screening. Chest X-ray, CT, and transesophageal echocardiography are available methods for the detection of aortic calcification [8]. A study led by Iijima et al. suggested that the grading of aortic calcifications based on chest X-rays could be valuable in the management and follow-up of aortic atherosclerosis because the detected aortic arch calcifications on a plain radiograph were prominent independent risk factors for cardiovascular events [9]. In a recent study, Woo et al. found that the aortic arch calcification seen on a conventional chest X-ray combined with coronary artery calcium score were beneficial in the detection and the prediction of outcome in patients suffering from angina [10].

Several twin studies have investigated the heritability of various aortic pathologies such as aortic aneurysms or stiffness [11,12,13]. However, currently there are no results on the heritability of aortic calcification and CTR, which could be important as an early predictor of cardiovascular disease progression.

The aim of this study was to investigate the contribution of genetic and environmental factors to cardiothoracic ratio and aortic calcification in our twin study population.

## 2. Materials and Methods

### 2.1. Subjects

In this study, we collected the chest X-rays of 342 twin pairs from South Korea in collaboration with the South Korean Twin Registry. Subjects over 18 years of age were included. Our exclusion criteria included pregnancy, breastfeeding, extreme obesity, history of cancer, and history of thoracic or cardiac surgery. We analyzed the PA chest X-rays of 261 MZ and 81 DZ twin pairs (male/female = 264/420). Cardiovascular risk factors and anthropometric data (height, weight, body mass index, waist circumference, left/right and mean systolic and diastolic blood pressures) were recorded. All subjects gave their informed consent for inclusion before they participated in the study. The study was conducted in accordance with the Declaration of Helsinki, and the protocol was approved by the Institutional Review Board of the Seoul Samsung Hospital (SMC 2005-08-113-049).

### 2.2. Measurement

The analysis of chest X-rays was performed using Adobe Illustrator on bitmap images converted from DICOM files. Cardiac width and thoracic width were measured as the largest distance between the left and right edges of the thorax and the heart. CTR was calculated by dividing the cardiac size by the thoracic size (Figure 1). A CTR above 50% was considered pathological. We used the scale described by Ogawa et al. for the analysis of aortic calcification (Figure 2 and Figure 3) [14]. The 16-part circle scale was placed on the chest X-ray over the aortic arch and the number of segments that showed signs of calcification were counted to determine the extent of calcification. The total number of calcified segments were divided by 16 to calculate the calcification score of the aortic arch. To classify the degree of the calcification, we defined the following four categories: no visible calcification, <50% calcification, >50% calcification, and circumferential calcification. We also categorized the calcifications on the basis of their appearance on the X-rays to the subgroups of no calcification, small spots or single thin lines of calcification, and one or more areas of thick calcification [9,15].

### 2.3. Statistical Analysis

The genetic and environmental contribution to cardiothoracic ratio and aortic calcification was determined using a classical twin study design. Monozygotic (MZ) twins share approximately 100% of their genes and dizygotic (DZ) twins share, on average, 50% of their segregating genes. The impact of genetic and environmental determinants of a given phenotype can be examined in twin studies. The ACE statistical model is the classical method used in twin studies to describe the phenotypic variance on the basis of the additive genetic component (A), common environmental component (C; shared by the twin pair such as age, socioeconomic status, etc.), and unique environmental components (E; factors affecting only one member of a twin pair such as diseases or smoking habit). The ACE statistical model was applied in order to examine the heritability of cardiac size, thoracic size, and CTR. Full ACE model and a series of sub-models were tested in each case. To determine the best fitting model with the least parameters, we used the Akaike information criteria (AIC). Due to the relatively healthy study population with limited number of subjects showing aortic calcification, we could not apply the ACE model in order to determine heritability of aortic arch calcification. Therefore, we investigated case-based comparisons of the discordant twins for aortic calcification. For the comparison of continuous variables between MZ and DZ twins, we applied independent samples *t*-test. Fishers’ exact test was used for dichotomous variables between the MZ and DZ twins. Statistical analysis was conducted using IBM SPSS and R with OpenMX package.

## 3. Results

### 3.1. Study Population

The descriptive characteristics of the study population (684 twins, mean age 38.6 ± 7.9 years) are summarized in Table 1. Significant differences between the MZ and DZ twins were shown regarding continuous variables such as age, height, weight, blood pressure, and thoracic width. The only statistically significant difference between the two groups as for dichotomous variables was hypertension. Most of our study population had no health issues. A few of our subjects had medical histories containing cardiovascular diseases such as dyslipidemia (*n* = 55), myocardial infarction (*n* = 9), stroke (*n* = 1), or hypertension (*n* = 63).

### 3.2. Cardiac Size, Thoracic Size, and Cardiothoracic Ratio

The ACE statistical model of cardiac size, thoracic size, and cardiothoracic ratio is summarized in Table 2. In the full ACE model of the thoracic width, the additive genetic component accounted for 0.67, while the unique environmental component accounted for 0.34. The common environmental component and age accounted for 0.0. The best fitting submodel was found to be the AE reduced model, resulting in the same outcome.

As seen in Table 2, in the full ACE model of the cardiac width the additive genetic component accounted for 0.42, the unique environmental component accounted for 0.12 while the common environmental component accounted for 0.46. The best fitting reduced submodel was found to be the AE model, resulting in the variance of 0.54 of additive genetic component and 0.46 of unique environmental factor.

In the full ACE model of the cardiothoracic ratio, the additive genetic component accounted for 0.46, the common environmental component and age accounted for 0.07, while the unique environmental component accounted for 0.47. Similar to the thoracic width and cardiac width, the best fitting submodel was found to be the AE model resulting in the variance of 0.54 of genetic component and 0.46 of unique environmental factors.

### 3.3. Aortic Arch Calcification of MZ and DZ Twins

As seen in Table 3, 8 MZ twin pairs were discordant for aortic calcification out of the total of 261 MZ twin pairs. Six out of eight pairs had calcifications categorized as small spots or single thin lines of calcification, while the other two pairs’ calcification fell in the subgroup of one or more areas of thick calcification. The subjects with the highest calcification score of 0.625 and 10/16 coverage also had aortic arch coverage of more than 50%. The lowest aortic calcification score was found to be 0.125 while covering 2/16 segments of the aortic arch. The results of aortic calcification in DZ twins are illustrated in Table 4. Similar to the MZ twins, there were only a small subset of pairs with detectable calcification. Four twin pairs out of the 81 pairs of DZ twins with only one member of each pair were affected. Moreover, the detected calcifications were covering less than 50% of the arch, and all fell in the subgroup of small spots or single thin lines of calcification. The highest aortic calcification score was found to be 0.313 while covering 5/16 segments of the aortic arch. The lowest aortic calcification score was 0.063 while covering only one segment of the aortic arch. We found more discordant twin pairs in the MZ group than in the DZ group, which suggests a rather higher environmental contribution to aortic calcification.

## 4. Discussion

The present study investigated the genetic and environmental contribution to cardiothoracic ratio and aortic arch calcification phenotypes. Our main results showed that in our study population, thoracic width, cardiac width, and cardiothoracic ratio had moderate heritability (ranging from 50 to 70% of individual variance). In our relatively young population, we did not find evidence for strong genetic influence on aortic arch calcification.

In this study, we conducted our research on PA chest X-rays of both MZ and DZ twin pairs while most studies in this topic used different imaging modalities and included family studies. The main advantage of our project compared to other published studies was the classical twin study design that allows for a better differentiation between genetic and environmental contribution. Our findings on cardiac width and size were consistent with several previous findings. Fagard et al. investigated the heritability of cardiac size using echocardiography and found that the genetic contributions had a statistically significant impact on the left ventricular mass. However, the results could depend on the inherited body size [16]. Another study led by Busjahn et al. investigated echocardiography on 166 twin pairs and found 59% genetic variance for left ventricular mass and septum thickness [17]. In another study, the same group used cardiac magnetic resonance imaging (CMR) on 25 twin pairs as it was proved to be a more accurate quantitative measurement of heart size compared to echocardiography [18,19]. They found even stronger evidence for the genetic influence on cardiac size than their previous echocardiographic study had revealed [20]. However, in the study of Adams et al., MZ and DZ twin pairs as well as siblings of the same sex all shared the same amount of intrapair variance in cardiac size even after 14 weeks of exercise training using echocardiography and electrocardiographic tests [21].

Our results on thoracic width heritability align with other study results. A family-based study by Williams-Blangero et al. showed a moderate genetic influence on thoracic width while investigating the heritability of chest dimensions in a high-altitude Tibetan population [22]. Another study led by Chatterjee et al. on 54 Indian twin pairs found that genetic factors play a larger role in determining chest circumference in comparison with environmental factors [23]. However, a few previous studies found a correlation between thoracic width and age. Mensah et al. measured the cardiothoracic ratio using chest radiographs similarly to our study. They found that transverse thoracic diameter increased parallel with age until approximately the sixth decade of life, from where it instead decreased with age [13]. As the mean age of our study population was significantly lower than theirs, age was not likely to have an effect on our results.

In this study, we also investigated potential genetic effects on aortic calcification using PA chest X-rays. Previous studies have found evidence of various different genetic variations playing a fundamental role in the development of arterial calcification [24]. A study by Rutsch et al. reviewed a wide range of genetic factors that have been associated with the development of aortic calcification and identified several specific genes [25]. Another study by O’Donell et al. found that abdominal aortic calcification detected on lateral lumbar radiographs was a heritable atherosclerotic trait [26]. In a twin study of 900 women, Cecelja et al. found that the phenotypic correlation between arterial stiffness and arterial calcification were attributed to a common genetic influence [27]. They also examined the heritability of aortic plaque and calcification of 100 twin women in another study and found high genetic contribution to aortic calcification [28].

In our current study, only a few twin pairs had detectable aortic calcification due to our relatively young sample. Furthermore, none of them included both members of a pair. However, a previous study found 36% heritability of aortic arch calcification detected on nonenhanced CT scans [29]. This could have been due to the several differences between our study concepts such as the different imaging modalities. Most studies focused on the abdominal aortic calcification detected on lateral lumbar radiographs or nonenhanced CT. In contrast to CT scans, conventional chest X-ray is a summation image that can only visualize the aortic arch and not the whole aorta [30].

Ogawa et al. found that the aortic calcification score detected on chest X-rays highly correlated with aortic arch calcification volume as compared to a multi-slice CT indicating that it is a suitable imaging modality for the purpose of this study [14]. The present study differs from existing literature that used chest X-rays for the detection of aortic calcification with regard to the fact that the previous studies were mainly conducted on patients with renal failure that were undergoing hemodialysis, a patient population that has more severe aortic calcifications that are thus easier to detect with chest X-rays [31]. Furthermore, the average age of the subject group was much higher in previous studies in comparison to ours [29]. However, our findings indicate no strong genetic contribution to aortic calcification in a younger age, which highlights the importance of unique environmental factors, such as healthy lifestyle, for the prevention of aortic atherosclerosis.

Our study has several limitations. In a study performed on healthy twins, Messerli et al. found that the sensitivity of chest X-ray for the detection of ascending aortic and aortic arch calcification was 52.3%, in contrast to low dose CT, which had a 96.2% sensitivity [32]. However, as healthy twins had no indications for CT scanning, it would have been ethically unacceptable to perform CT for the purpose of this study, which is why chest X-rays were obtained instead. Although other imaging modalities such as cardiac magnetic resonance imaging allow for more accurate quantitative measurements, it is not as easily accessible and available as the chest radiographs used in the present study [18,19]. Performing a detailed heritability analysis of the aortic arch calcification was not possible due to the small population of twins with detectable aortic calcification on chest X-ray. We also have to take into account that that the majority of the existing literature has indicated that the genetic component plays a moderately large role in the development of aortic calcifications. For this purpose, future studies may employ a population group with a higher average age than ours, as previous studies with an even smaller twin pair sample size have detected a larger number of aortic calcifications than our study [29,33,34]. In comparison with most of the existing literature, our study involved 342 twin pairs of South Korean origin. It could be possible that the reason for the limited amount of detectable aortic calcification were due to the lower general prevalence of aortic calcification in the Asian population compared to Caucasian population. This theory is supported by a study lead by El-Saed et al. on Japanese men, Caucasian men, and Japanese-American men that compared the prevalence of aortic calcification after adjusting for smoking habits and other known risk factors. They analyzed the presence of aortic calcification by electron beam tomography on approximately 300 men aged 40–49 from each subject group. They found that Japanese men had a significantly lower presence of aortic calcification as compared to the two other groups, while the Caucasian men had the highest prevalence [35]. However, their study only included males, while in our study we investigated both male and female patients as the sex of the patient could affect the results. In fact, in our study, the significant differences between the MZ and DZ twins regarding continuous variables (age, height, weight, blood pressure, and thoracic width) could be a result of the higher female-to-male ratio in the MZ group. Thus, future studies should be performed in different ethnic groups and ideally adjusted on the basis of the prevalence of aortic calcification for that specific ethnic group.

## 5. Conclusions

In summary, we found strong heritability for thoracic width and moderate heritability for cardiac width and cardiothoracic ratio. In this relatively young population, our results indicated no substantial genetic contribution to aortic calcification detected on PA chest X-rays. However, further studies on larger and older twin populations are warranted to draw clinically significant conclusions.

## Figures and Tables

**Figure 1 medicina-57-00421-f001:**
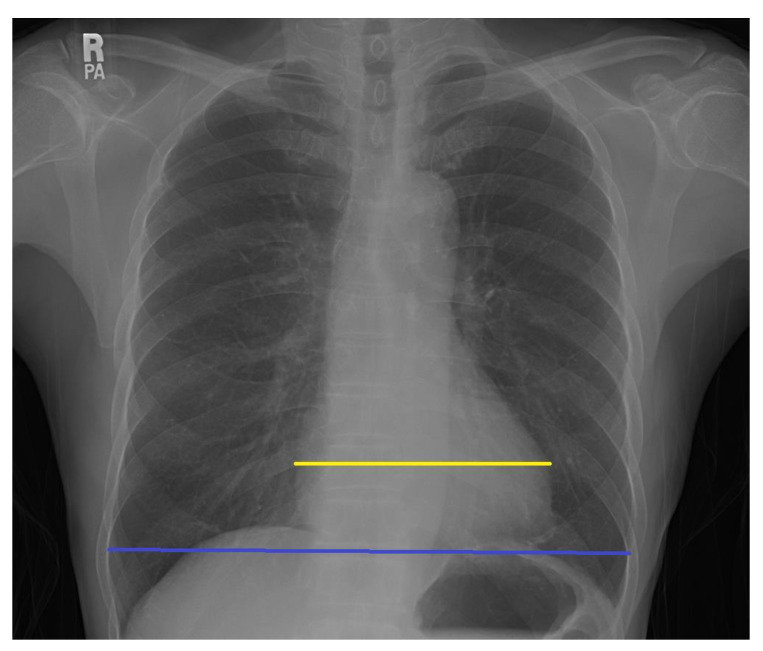
Measurement of the cardiac width (yellow line) and thoracic width (blue line). Cardiothoracic ratio is calculated by dividing the cardiac width with the thoracic width.

**Figure 2 medicina-57-00421-f002:**
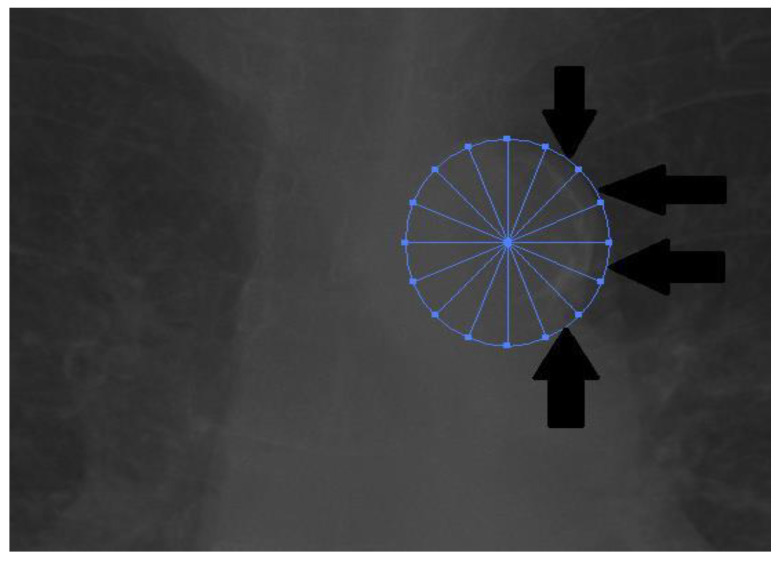
The 16-part circle (with 16 radial dividers and 0 concentric dividers) placed above the aortic arch as seen on PA chest X-ray.

**Figure 3 medicina-57-00421-f003:**
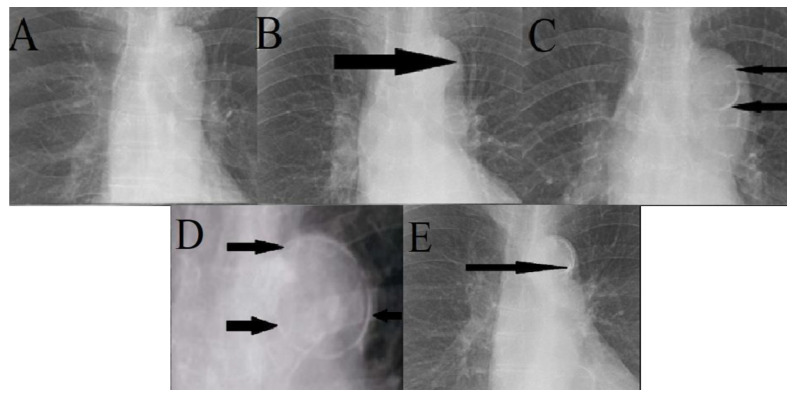
(**A**) No aortic arch calcification visible on PA chest X-ray (Grade 0). (**B**) Less than 50% aortic arch calcification categorized as small spot or single thin line of calcification as seen on PA chest X-ray (Grade 1). (**C**) Aortic arch calcification with more than 50% calcification and graded as small spots or single thin line of calcification as seen on PA chest X-ray (Grade 2). (**D**) Aortic arch calcification graded as a circumferential calcification as seen on PA chest X-ray (Grade 3). (**E**) Aortic arch calcification graded as one or more areas of thick calcification as seen on PA chest X-ray.

**Table 1 medicina-57-00421-t001:** Demographics and descriptive clinical characteristics of the study subjects. The continuous variables are expressed as the mean ± standard deviation, and corresponding *p*-values are calculated by independent-samples *t*-test. The dichotomous variables are expressed as *n* (%), and corresponding *p*-values are calculated by Fisher’s exact test. * = *p* < 0.1, ** = *p* < 0.05, *** = *p* < 0.01.

	Total (*n* = 684)	MZ Twins (*n* = 522)	DZ Twins (*n* = 162)
Age (years)	38.6 ± 8.0	38.3 ** ± 8.0	39.8 ** ± 7.7
Sex	Males: 264 (39%)	Males: 192 (37%)	Males: 72 (44%)
	Females: 420 (61%)	Females: 330 (63%)	Females: 90 (56%)
Height (cm)	162.2 ± 8.3	161.9 * ± 8.1	163.2 * ± 8.7
Weight (kg)	61.3 ± 11.7	60.8 * ± 11.2	62.9 * ± 12.9
Waist (cm)	79.0 ± 8.9	78.8 ± 8.6	79.8 ± 9.5
BMI (kg/m^2^)	23.2 ± 3.1	23.1 ± 3.0	23.4 ± 3.4
Systolic blood pressure, left (mmHg)	109.5 ± 15.9	107.8 *** ± 15.6	115.0 *** ± 15.6
Diastolic blood pressure, left (mmHg)	69.4 ± 10.7	68.5 *** ± 10.9	72.2 *** ± 9.4
Systolic blood pressure, right (mmHg)	109.7 ± 16.0	108.0 *** ± 15.5	115.4 *** ± 16.1
Diastolic blood pressure, right (mmHg)	69.4 ± 10.7	68.5 *** ± 10.9	72.1 *** ± 9.8
Mean systolic blood pressure (mmHg)	111.0 ± 16.1	109.2 *** ± 15.8	116.8 *** ± 16.0
Mean diastolic blood pressure (mmHg)	70.2 ± 11.0	69.3 *** ± 11.1	73.1 *** ± 9.9
Smoking *n* (%)	241 (35%)	177 (33%)	64 (40%)
History of stroke, *n* (%)	1 (0.1%)	1 (0.2%)	0 (0%)
History of myocardial infarction, *n* (%)	9 (1%)	5 (1%)	4 (2%)
Hypertension, *n* (%)	63 (9%)	40 (8%)	23 (14%)
Hyperlipidemia, *n* (%)	55 (8%)	42 (8%)	13 (8%)
Diabetes mellitus, *n* (%)	22 (3%)	18 (3%)	4 (2%)
Cardiac width (cm)	22.9 ± 3.1	22.8 ± 3.0	23.0 ± 3.4
Thoracic width (cm)	50.0 ± 5.2	49.8 * ± 4.8	50.8 * ± 6.3
Cardiothoracic ratio	0.5 ± 0.1	0.5 ± 0.1	0.5 ± 0.1
No visible calcification	0.98 ± 0.13	0.98 ± 0.12	0.98 ± 0.16
Small spots or single thin area	0.01 ± 0.12	0.01 ± 0.11	0.02 ± 0.16
One or more areas of thick calcification	0 ± 0.05	0 ± 0.06	0 ± 0
Circular calcification	0 ± 0	0 ± 0	0 ± 0
No visible calcification_2	0.98 ± 0.13	0.98 ± 0.12	0.98 ± 0.16
<50% calc_2	0.02 ± 0.13	0.01 ± 0.11	0.02 ± 0.16
>50% calc_2	0 ± 0.04	0 ± 0.04	0 ± 0
Circular calcification_2	0 ± 0	0 ± 0	0 ± 0
Number of calcificated segments	0.07 ± 0.58	0.06 ± 0.59	0.08 ± 0.56
Aortic calcification score	0 ± 0.04	0 ± 0.04	0.01 ± 0.03

**Table 2 medicina-57-00421-t002:** Full ACE statistical model and reduced submodels for the cardiac width (ACE model 1), thoracic width (ACE model 2), and cardiothoracic ratio (ACE model 3). Number of twin pairs: 261 MZ, 81 DZ. Best fitting model is marked with asterisk (*).

ACE model 1							Results (95% CI)		
**Thoracic width**	AIC	BIC	-2LL	df	DiffLL	*p* value	A	C	E
ACE	3171.247	3175.550	3156.912	7	Reference	Reference	0.67 (0.39, 0.73)	0 (0, 0.261)	0.33 (0.28, 0.41)
**AE ***	3169.162	3172.887	3156.912	6	0	1	0.67 (0.59, 0.73)	0	0.33 (0.28, 0.41)
CE	3190.850	3194.575	3178.599	6	−21.688	0	0	0.57 (0.49, 0.64)	0.43 (0.36, 0.51)
E	3320.370	3323.504	3310.191	5	−153.28	0	0	0	1
SAT	3175.126	3181.087	Rmz: 0.66 (0.585 0.724)	Rdz: 0.307 (0.092 0.495)					
**ACE model 2**							**Results (95% CI)**		
**Cardiac width**	AIC	BIC	-2LL	df	DiffLL	*p* value	A	C	E
ACE	2939.857	2944.159	2925.521	7	Reference	Reference	0.42 (0.05, 0.62)	0.12 (0, 0.459)	0.46 (0.38, 0.56)
**AE***	2938.092	2941.817	2925.841	6	−0.32	0.572	0.54 (0.45, 0.62)	0	0.46 (0.38, 0.55)
CE	2942.754	2946.479	2930.504	6	−4.982	0.026	0	0.49 (0.40, 0.56)	0.51 (0.44, 0.60)
E	3026.992	3030.127	3016.814	5	−91.292	0	0	0	1
SAT	2945.265	2951.227	Rmz: 0.537 (0.442 0.62)	Rdz: 0.324 (0.113 0.506)					
**ACE model 3**							**Results (95% CI)**		
**Cardiothoracic ratio**	AIC	BIC	-2LL	df	DiffLL	*p* value	A	C	E
ACE	−2400	−2395.697	−2414.335	7	Reference	Reference	0.46 (0.18, 0.61)	0.07 (0, 0.38)	0.47 (0.39, 0.56)
**AE***	−2401.937	−2398.212	−2414.188	6	−0.148	0.701	0.54 (0.44, 0.62)	0	0.46 (0.39, 0.56)
CE	−2395.032	−2391.307	−2407.283	6	−7.052	0.008	0	0.46 (0.37, 0.54)	0.54 (0.46, 0.63)
E	−2316.544	−2313.410	−2326.722	5	−87.613	0	0	0	1
SAT	−2399.927	−2393.966	Rmz: 0.508 (0.412 0.593)	Rdz: 0.348 (0.142 0.525)					

AIC: Akaike information criterion, BIC: Bayesian information criterion, -2LL: 2 × loglikelihood value of the model, df: degree of freedom of the model, DiffLL: difference between -2LL of the model compared to -2LL of the reference (ACE) model, *p*-value: *p*-value calculated on the basis of DiffLL and chi-squared distribution, rDZ: dizygotic intra-pair correlation coefficient, rMZ: monozygotic intra-pair correlation coefficient.

**Table 3 medicina-57-00421-t003:** The aortic calcification parameters of the MZ twin pairs where at least one member of the twin pair had an aortic calcification detectable on PA chest X-ray.

MZ Twin-Pairs	Type of Calcification		Percentage of Calcification (%)		Number of Calcified Segments		Aortic Calcification Score	
Pair 1	Twin A: Small spots or a single, thin area of calcification	Twin B: No visible calcification	Twin A: <50	Twin B: 0	Twin A: 2	Twin B: 0	Twin A: 0.125	Twin B: 0
Pair 2	Twin A: Small spots or a single, thin area of calcification	Twin B: No visible calcification	Twin A: <50	Twin B: 0	Twin A: 5	Twin B: 0	Twin A: 0.313	Twin B: 0
Pair 3	Twin A: One or more areas of thick calcification	Twin B: No visible calcification	Twin A: >50	Twin B: 0	Twin A: 10	Twin B: 0	Twin A: 0.625	Twin B: 0
Pair 4	Twin A: One or more areas of thick calcification	Twin B: No visible calcification	Twin A: <50	Twin B: 0	Twin A: 2	Twin B: 0	Twin A: 0.125	Twin B: 0
Pair 5	Twin A: Small spots or a single, thin area of calcification	Twin B: No visible calcification	Twin A: <50	Twin B: 0	Twin A: 3	Twin B: 0	Twin A: 0.188	Twin B: 0
Pair 6	Twin A: Small spots or a single, thin area of calcification	Twin B: No visible calcification	Twin A: <50	Twin B: 0	Twin A: 3	Twin B: 0	Twin A: 0.188	Twin B: 0
Pair 7	Twin A: Small spots or a single, thin area of calcification	Twin B: No visible calcification	Twin A: <50	Twin B: 0	Twin A: 4	Twin B: 0	Twin A: 0.250	Twin B: 0
Pair 8	Twin A: Small spots or a single, thin area of calcification	Twin B: No visible calcification	Twin A: <50	Twin B: 0	Twin A: 4	Twin B: 0	Twin A: 0.250	Twin B: 0

**Table 4 medicina-57-00421-t004:** The aortic calcification parameters of the DZ twin pairs where at least one of the members of the twin pair had an aortic calcification detectable on PA chest X-ray.

DZ Twin-Pairs	Type of Calcification		Percentage of Calcification (%)		Number of Calcified Segments		Aortic Calcification Score	
Pair 1	Twin A: Small spots or a single, thin area of calcification	Twin B: No visible calcification	Twin A: <50	Twin B: 0	Twin A: 3	Twin B: 0	Twin A: 0.188	Twin B: 0
Pair 2	Twin A: Small spots or a single, thin area of calcification	Twin B: No visible calcification	Twin A: <50	Twin B: 0	Twin A: 4	Twin B: 0	Twin A: 0.250	Twin B: 0
Pair 3	Twin A: Small spots or a single, thin area of calcification	Twin B: No visible calcification	Twin A: <50	Twin B: 0	Twin A: 5	Twin B: 0	Twin A: 0.313	Twin B: 0
Pair 4	Twin A: Small spots or a single, thin area of calcification	Twin B: No visible calcification	Twin A: <50	Twin B: 0	Twin A: 1	Twin B: 0	Twin A: 0.063	Twin B: 0

## Data Availability

Data supporting reported results can be provided upon request.

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
