# Peer review of "Heritability of Cardiothoracic Ratio and Aortic Arch Calcification in Twins"

_medicina, 2021, doi:10.3390/medicina57050421_

Round 1

Reviewer 1 Report

Thank you for the review.
The authors present a study that  concludes that the heritable component was significant regarding thoracic width, cardiac width and the CTR.

The project is well thought. The methodology is clear and results are presented in a clear way. I would refrain from commenting on statistical analysis as that is not my expertise. The discussion is well written , and referenced. The language used is appropriate .

I would recommend publication if the statistical analysis is found appropriate.

Thank you 

Reviewer 2 Report

The authors have corrected the manuscript as suggested.   

This manuscript is a resubmission of an earlier submission. The following is a list of the peer review reports and author responses from that submission.

Round 1

Reviewer 1 Report

The authors present Aortic arch calcification (AoAC) and association with a variety of cardiovascular complications by investigating the quantitative contribution of genetics and environments to the AoAC and CTR where they studied a total of 686 twins from the South-Korean twin registry and recorded  PA chest X- rays, and their cardiovascular risk factors and anthropometric data . The authors found a detectable AoAC  for only few subjects in this population. They found the thoracic width had strong heritability, cardiac width was determined moderately by genetic factors and the CTR also had a moderate degree of heritability and they concluded that the heritable component was significant regarding thoracic width, cardiac width and the CTR.

The paper has good clinical significance and is well written . I would suggest getting it reviewed by a native English speaker .

The manuscript is otherwise well written. 

Reviewer 2 Report

This study estimates heritability for thoratic width, cardiac width and cardiothoratic ratio based on Korean twin data. The study is interesting since it is based on X-ray scanning and the heritability estimates of these anatomic traits are not well known. Further, East Asian populations are generally much less studied than the USA and European populations thus warranting more researcher. I have only minor comments.

Abstract: “… AoAC and CTR are not …” (currently “is not”).

Abstracts and main text: I think that it is enough to give only two decimals for A, C and E estimates (this is also unsystematic since in some cases two and other cases three decimals are given).

Sex ratio should be given in Methods section.

Statistical methods (page 4): ”… share nearly 50% of their genes…” should read “share, on average, 50% of their segregating genes …”

Table 1: When the sample size is 686, it does not make sense to give decimals for percentages.

Table 1: The way how percentages are calculated is strange. For example, in smoking it shows % of MZ and DZ twins of smokers. I think that it would be much more informative to give % of smokers in these cells (all, MZ and DZ).

Table 2 is difficult to read. I would present the model fit statistics as a separate table and so A, C and E estimates with 95% CIs can be presented in own columns making the table easier to read.

Reviewer 3 Report

Dear authors, I have reviewed your manuscript entitled: Heritability of cardiothoracic ratio and aortic arch calcification in twins and Ihave the following comments:

  1. I recommend to provide the BMI value in Table 1.
  2. I recommend to re-design Table 2, since it is completely unreadable. I cannot see P-values.
  3. In abstract please add statistical values.
  4. The major limitation of this study is lack of analysis of fasting blood glucose, HbA1c and the level of C-reactive protein. The authors should include the analysis of those parameters to the manuscript.
  5. In the materials and methods sections, please provide a table to summarize the inclusion and exclusion criteria, including age range, BMI…..
  6. Minor modification of the grammar is required.
  7. The font and its size should be uniform in entire manuscript.
  8. Abbreviations should be placed under the table, not in the description of table.
  9. General: interesting, well conducted work.